# How the public imagines the energy future: Exploring and clustering non-experts' techno-economic expectations towards the future energy system

Lukas Braunreiter[1,2]*, Michael Stauffacher[2], Yann Benedict Blumer[1]

**1** Institute for Innovation & Entrepreneurship, School of Management and Law, ZHAW, Winterthur, Switzerland, **2** ETH Zurich, Institute for Environmental Decisions (IED), Transdisciplinarity Laboratory (TdLab), Zurich, Switzerland

* lukas.braunreiter@zhaw.ch

**Data Availability Statement:** The raw data set is available at the Zenodo repository: 10.5281/zenodo.3395054.

## Abstract

Various countries have pledged to carry out system-wide energy transitions to address climate change. This requires taking strategic decisions with long-term consequences under conditions of considerable uncertainty. For this reason, many actors in the energy sector develop model-based scenarios to guide debates and decision-making about plausible future energy systems. Besides being a decision support instrument for policy-makers, energy scenarios are widely recognized as a way of shaping the expectations of experts and of influencing energy policy more generally. However, relatively little is known about how energy scenarios shape preferences and expectations of the public. We use an explorative research design to assess the publics' expectations of future energy systems through an online survey among Swiss residents (N = 797). We identified four significantly different clusters of people with distinct expectations about the future energy system, each seeing different implications for the acceptability of energy policies and the compatibility with projections of techno-economic energy scenarios. Cluster 1 expects a system-wide energy transition towards renewable energy sources that is similar to the policy-relevant national energy scenario. Cluster 2 also expects an energy transition, but believes it will lead to a range of technical challenges, societal conflicts and controversies with neighboring countries. Cluster 3 is the only cluster not expecting significant changes in the future energy system and thus not anticipating an energy transition. Cluster 4's expectations are between cluster 1 and 2, but it anticipates a huge increase in per capita electricity demand while prices are expected to remain low. The study at hand offers some initial insights into the interdependencies between energy transition pathways outlined in techno-economic energy scenarios and the energy system expectations of the public. These insights are essential for gaining a better understanding of whether and how energy scenarios can contribute to informed public debates about energy futures and how desirable pathways towards them might look like.

**Funding:** This work has been supported by the Swiss Competence Center for Research in Energy, Society and Transition (SCCER CREST, Innosuisse Grant No. 1155000154) to LB and YBB. The funder had no role in study design, data collection and analysis, decision to publish, or preparation of the manuscript.

**Competing interests:** The authors have declared that no competing interests exist.

# Introduction

## 1.1 Energy scenarios & public discourse

Various countries have pledged to carry out system-wide energy transitions to address climate change [1]. This requires taking strategic decisions with long-term consequences under conditions of considerable uncertainty [2]. For this reason, many countries develop or commission scenarios consisting of plausible pathways for a system change without impairing an affordable and reliable energy supply [3, 4].

Scenarios support decisions in many contexts and are typically developed and applied by academic, corporate, or governmental communities of experts [5, 6]. The energy sector has a long history of predominantly normative and proprietary scenario use [7, 8]. Today, public and private actors often publish the results of scenario studies to legitimize decisions, increase the transparency of decision-making, and to direct energy policy debate towards a particular vision of the energy future [9]. These energy scenarios exceed the typical time horizon of political processes by extending to the year 2050 and beyond, thereby implying the relevance of the socio-technical configuration of the energy system in the distant future as a basis for contemporary planning [10].

How scenarios shape understanding and support of specific policies has so far mostly been studied in the context of experts [11, 12]. At the same time, relatively little is known about the influence of energy scenarios on the public. This would, however, be important, as Demski et al. [13] have demonstrated that the pathways of energy scenarios function as powerful framing object for individual opinion formation and energy technology preferences. Some scholars, for example [14] have made first attempts to describe the influence of energy scenarios on non-experts. Recent findings by [15] furthermore suggest that interactive web-tools are not more efficient in communicating scenario content than conventional storylines. Most studies focusing on non-experts have in common that they scrutinize the influence of scenarios under experimental conditions. They test how participants react to scenario products and observe what short- and long-term effects these have on opinion formation processes. In reality, however, the public cannot be expected to actively and consciously consult or use energy scenarios. Instead, they receive scenario-based insights indirectly and in a fragmented way, for example via the media or political discussions, often in the form of specific promises and concerns. This is why our exploratory approach intends to assess the publics' energy system expectations without triggering them by scenarios or other forms of energy visions.

## 1.2 Expectations are vital for understanding individual perceptions of the future

How individuals conceptualise the future energy system outside of a lab setting is not yet understood very well. In general, individuals' views of the future can be conceptualized as expectations. Expectations are informal and often partially held beliefs about the future [16, 17]. Expectations may be entirely personal and tacit commitments to a future possibility. They can influence how people integrate new information and hence develop particular attitudes [14, 18, 19]. As [20] stated, behavior and decision-making in the present are anchored in the perception of the future.

Public acceptance studies tend to focus on stated preferences and beliefs that typically exclude expectations or perceptions of the future. This stands in contrast to the uncertainty and long-term focus inherent in the idea of an energy transition and the related goal of mitigating climate change. In fact, people differ both in their perception of long timeframes [21] as well as in their consideration of future consequences [22]. With the exception of [23], previous

studies on public expectations have rarely focused on a specific socio-technical system in an in-depth way and only first attempts to create scales for assessing expectations in the energy system exist [24, 25]. Furthermore, there is no substantial body of research trying to explore and classify expectations of the future energy system among the public.

## 1.3 Collectively held expectations

If expectations are collectively held, they shape a shared understanding between actors that can ultimately become a normative force [26]. A range of case studies show that if relevant decision-makers all share the same expectations, this can impact the development and diffusion of novel technologies in otherwise relatively stable socio-technical systems [26–29]. If widely shared, expectations become publicly held visions of a desirable future [30]. At this point, expectations are no longer personal and tacit but become a performative power, influencing present-day behavior [26]. This self-fulfilling dimension can shape infrastructures and institutions, linking collectively held expectations to policy and politics, which is why the public energy system expectations matter [28, 31].

Assessing the public's collectively held energy system expectations provides insights into the potential social opportunities and constraints of techno-economic energy pathways that typically neglect societal perspectives [13]. This is important because the public is a crucial actor in energy transitions, with various roles that include accepting energy infrastructure, supporting energy policies, adapting energy demands, or adopting energy technologies [32]. The public's energy system expectations influence how likely, acceptable, or desirable alternative energy futures appear [30, 31, 33].

At the same time, an assessment of the public's energy system expectations indicates how strongly energy scenarios function as framing lenses in the energy discourse [34]. In the sense of Grunwald [9], this enables a better understanding of whether and how energy scenarios enlighten public debate by aligning their energy system expectations with the values, assumptions, and interests represented in techno-economic energy scenarios. While techno-economic energy scenarios may shape energy system expectations, techno-economic energy scenarios are, in turn, also influenced by the public's energy system expectations. Fundamentally, energy scenarios are social constructs based on assumptions and values that are contingent on the society in which they are formed. Ellenbeck & Lilliestam [35], for example, demonstrated that energy models and assumptions reflect the scenario developers' understanding of society and thus reproduce particular expert discourses.

## 1.4 Study aim

In this study, we explore public perception of a range of particular promises and concerns about the future energy system, which we refer to as *expectations*. To operationalize expectations, we conducted a survey among a sample of Swiss residents. In particular, we address the following research questions: i) What are the public's expectations about the techno-economic development of the energy system and how stable are they in the face of different time horizons and framings? ii) Are there different types of expectations towards the future energy system that can be identified among the public? iii) How do the public's energy system expectations relate to projections made in the policy-relevant energy scenario? In this way, our socio-scientific perspective provides empirical evidence of interdependencies between the formalized projections of techno-economic energy scenarios and the informal expectations of the energy future among the public.

## Methods and procedure

### 2.1 Ethics statement

Data was collected from an online survey. Participation in the survey was voluntary. At the beginning of this survey, participants' were informed in written form that the responses they provided were going to be used for research purposes only. Furthermore, they were informed that the data was going to be analyzed and published in an anonymous form. This kind of non-invasive research does not require approval of an ethics committee according to the Swiss Federal Act on Research Involving Human Beings [36].

### 2.2 Case description: The relevance of energy scenarios to Swiss energy strategy 2050

Switzerland is an example of an industrialized country with a distinctive mix of energy sources and uses. Although not a member of the European Union, Switzerland is nevertheless very much integrated with international energy markets [37, 38]. We chose to survey residents in Switzerland because the country represents an ideal case for an empirical study of public energy system expectations and their alignment with scenario-derived energy policy. This is for a number of reasons. First, the nation's direct-democratic system allows the population to decide on a range of particular political issues, including energy policy. The most recent example was in 2017 when Swiss citizens enacted the Energy Strategy 2050 (ES2050) through a popular referendum. Hence, a significant share of the population is familiar with energy policy-related promises and concerns, and even the lengthy planning timeframes associated with an energy transition. Second, techno-economic energy scenarios were instrumental in the development of ES2050. In the aftermath of the Fukushima accident in 2011, the Swiss government decided to phase-out nuclear power gradually, although that it still generates about one-third of the nation's power supply. To identify cost-efficient and technically feasible ways of achieving this phase-out, the Swiss Federal Office of Energy (SFOE) developed a scenario study that subsequently functioned as an information basis for the development of ES2050. The respective scenario studies explores three different futures for the Swiss energy system. While scenarios ought to consider multiple futures without attaching probabilities from a methodological perspective, the policymaking processes reduced this plurality to a single pathway that ultimately was the basis for ES2050. This is why, from a voter perspective, ES2050 was presented as a single set of energy policy measures and targets. A range of scenario-based projections, for example, related to the cost of the proposed transition or its effects on the nation's reliance on electricity imports, were discussed at length in the political campaign leading up to the ES2050 referendum [38, 39].

### 2.3 Sample description

Data collection took place in December 2017. Survey participants were recruited via an online panel. Panel members received an invitation to participate in the study, with a small incentive of about 0.75 Euro credited upon completion of the survey.

The data analyzed here is part of a larger online survey covering a broad spectrum of energy-related attitudes. Detailed descriptions of the questionnaire development and participant recruitment process can be found in [18]. We applied quota sampling for the categories age and gender. In particular, five age categories (18–29; 30–39; 40–49; 50–59, 60+) were defined per gender. Once a quota was filled, additional respondents belonging to the category were screened out at the beginning of the survey. In total, 806 German-speaking respondents completed the survey of which 797 provided useful answers. 35 participants were screened out.

640 participants form the main sample, and 157 participants are part of an experimental group. The experimental group completed the same questionnaire as the main sample, but were given a different framing or time horizon for selected questions. These differences are presented in detail in section 2.3. There are no significant differences between the main sample and the experimental sample in terms of age, gender, political orientation and education. The samples is representative of the Swiss population in terms of age, gender, and political party identification (see S1 Appendix). The share of university degree holders is slightly lower in the sample, 22.7% in the sample as opposed to 27% in the Swiss population. In addition, the assessed energy technology preferences are in line with recent attitude surveys among residents of Switzerland [40, 41].

## 2.4 Questionnaire: Items used in this study

Out of the longer questionnaire used for the survey, four question blocks have been analyzed in detail for this study:

The first contained questions on general energy issues: This includes the preference for renewable and non-renewable energy technologies, the perceived need for an energy transition, and the preference for locally generated electricity.

The second contained items operationalizing energy system expectations. The key rationale was to include items that in combination provide a meaningful description of the critical dimensions of the future energy system. In total, ten distinct energy system expectations were included (see Table 1). They were based on Gregorowius & Beuttler [25] and adapted by Blumer et al. [18]. In the latter study, expectation items were not explored in detail but aggregated: six of them were used in a larger regression analysis focusing on the acceptability of hydropower and deep geothermal energy. While some items describe the extent of the energy transition (for example the share of renewables), others describe the state of the energy system (for example the prevalence of power outages) or potential areas of conflict (for example related to

**Table 1. Energy system expectations for the year 2050 of the main sample (N = 640).**

| *How do you expect [item] to change by 2050?* | *M* | *SD* |
|---|---|---|
| **TransitionExtent items** | | |
| Renewables | 5.59 | 1.12 |
| Energy efficiency | 5.63 | 1.13 |
| Electric vehicles | 5.54 | 1.16 |
| **SystemState items** | | |
| Electricity use per capita | 4.61 | 1.33 |
| Oil and gas prices | 5.21 | 1.29 |
| Electricity prices | 4.80 | 1.21 |
| Imported electricity | 4.18 | 1.20 |
| Power outages | 3.88 | 1.22 |
| Societal conflicts over energy infrastructure | 4.60 | 1.20 |
| Energy related controversies with neighboring countries | 4.44 | 1.09 |

*Notes.* Overview of energy system expectation of the main sample (n = 640) for the year 2050. M = Mean, SD = Standard Deviation. Survey participants were provided with a seven-point scale for each item to indicate how they expect it to develop in comparison to today. The middle of the scale corresponds to a situation like today (e.g., 4 = share of electric vehicles is expected to remain the same), whereas the endpoints would refer to a sharp increase (7) or decrease (1). The subdivisions *TransitionExtent*, describing the scale of the energy transition, and *SystemState*, describing the conditions of the future energy system are the result of a factor analysis (see S1 Appendix).

energy infrastructure). All items describe energy system characteristics that are typically projected—be it explicit or implicit—in techno-economic energy scenarios. Survey participants were asked to indicate how they expected these characteristics to have changed in relative terms by the year 2050 (2030 for the experimental group) on a slider bar ranging from one (*sharp decrease*) to seven (*sharp increase*) with a starting position of four (*same as today*). The year 2050 was chosen as it is the reference year for the Swiss ES2050 as well as a standard reference year for climate and energy-related strategies. The 2030 timeframe used for the experimental group was chosen because it is far enough in the future for changes in the energy system to happen, but close enough for survey participants to imagine and significantly closer to the present than 2050.

The third block contained a task in which participants were asked to estimate the absolute share of renewables in the energy mix in 2050. For that purpose, we provided the latest historic share of 2016 (21%) as a reference point and respondents could indicate their estimation for 2050 on a slider bar from 0 to 100 percent. While the main sample got an idealistic framing ("According to your own values and preferences, how high should the share of renewables be in 2050?"), the experimental sample got a realistic framing ("Considering economic and political realities, what do you think the share of renewables will be in 2050?").

The fourth block contained a set of items to assess respondent's political ideology, trust in institutions and science, as well as their future orientation, using the 12-item Consideration of Future Consequences scale (Joireman et al., 2008).

The survey ended with a set of demographic questions. Throughout the survey, we used a seven-point Likert scale ranging from 1 (*totally disagree*) to 7 (*totally agree*), if it is not stated otherwise. On average, respondents required 16.3 minutes to complete the survey, and 90% of respondents were able to finish within 31 minutes.

## 2.5 Data analysis

Data was analyzed using the IBM SPSS software package (version 25). For research question i), descriptive statistics and a factor analysis were conducted, with the latter suggesting that two dimensions can represent the energy system expectations accurately. The first dimension, *Transition Extent*, is composed of three energy system expectation items describing the degree to which the energy system has transitioned (i.e. share of renewables, share of electric vehicles and the efficiency of appliances & processes). The second dimension, *System State*, consists of the remaining seven energy system expectations that address potential challenges and conflicts associated with the future energy system (i.e. likelihood of controversies with neighboring countries, power outages or increasing energy prices). Both dimensions have a good reliability score (for details, see S1 Appendix.

To identify patterns in the participants' energy system expectations (research question ii), a hierarchical cluster analysis (Ward method with squared Euclidian distance) [42] was applied to the main sample. Ward's minimum variance criterion minimizes the total within-cluster variance. To achieve this, at each stage, the pair of clusters that leads to a minimum increase in total within-cluster variance after merging is identified [43]. Examination of the cluster coefficients suggests that three, four, or five cluster solutions are conceivable.

Further data analysis by the authors showed that a three-cluster solution generates two almost identical clusters that makes the interpretation of the data very difficult, and a five-cluster solution creates vastly uneven cluster sizes, This is why reporting results for clustering solutions with 4 clusters was preferred. To clarify the procedure, we present the steps from the three- to the four- and five-cluster solution (see S1 Appendix). Because cluster analysis can be sensitive to the ordering of cases, several analyses with differing case sequences were

conducted. While the case numbers differ slightly, the significant differences with respect to the expectation variables produce a stable pattern in all those solutions.

We then performed an analysis of variance (ANOVA) to test for significant differences between the clusters and the respondents' attitudes about energy in general and sociodemographic data, i.e. question block one and four. In general, we used Bonferroni as *post hoc* tests for statistical significance, which controls for the multiple number of comparisons by dividing through the total number of tests. However, because Levene's test of homogeneity of variance is significant for some of the dependent variables (both in the socio-demographic and the energy attitude ANOVA, (p <0.05) and the cluster sizes are unequal, we also used Games-Howell as *post hoc* tests for statistical significance [43]. The ANOVA shows that the four clusters differ in their acceptance of energy technologies, support for the national energy strategy, trust in political institutions and science, future orientation, and demographic background. A second ANOVA was performed to demonstrate the relationship between the relative scores of the energy system expectations and the absolute values which respondents ascribe to renewables in the future in Part 3 of the questionnaire.

For research question iii), which relates the clusters' energy system expectations with the energy scenario "*Energy Perspectives*" that forms the scientific basis for ES2050, a content analysis of the 900-page scenario study was conducted [44]. For most energy system expectations used in the study, a corresponding scenario projection can be found, even though some of them are only implicitly considered. For example, acceptability is often only represented through the underlying potential ascribed to certain technologies and it is not in all cases transparent what particular assumptions were made by modellers. Exemplifying this is the case of hydropower: Switzerland has a long history of hydropower use. While the mountainous regions would offer many more opportunities with suitable geophysical properties for hydropower plants, additional reservoirs would with few exceptions require the flooding of inhabited valleys or pristine ecological environments. Hence, the limited potential ascribed to new hydropower plants in "*Energy Perspectives*" reflects the strong implicit assumptions about its social acceptance. After the explicit and implicit scenario projections corresponding with the public energy system expectations were identified, the authors of this paper rated their fit. A simple three part rating system was applied that labeled the fit between the scenario projection and the public's expectation either as *close*, *average*, or *distant* was applied. While some ratings were unequivocal (e.g. cluster expect strong increase, scenario projection a decrease), the comparison between the scenario projection (typically absolute values) and the public's expectations (relative to today) was sometimes challenging. Nevertheless, we opted for this direct way of comparison to be able to highlight both the evident similarities and the striking mismatches between the two conceptualizations of the energy future.

## Results

### 3.1 The public's energy system expectations

Respondents from the main sample (n = 640) overwhelmingly expect the energy system to have changed significantly by 2050 (see Table 1). The most substantial changes from the status quo (i.e., represented by a value of 4) are in the increased share of renewables (M = 5.59), the increased energy efficiency of appliances and processes (M = 5.63), and a larger number of electric vehicles in the passenger car fleet (M = 5.54). The results also show that the public expects oil prices (M = 5.21) to increase more than electricity prices (M = 4.80) and the per capita consumption of electricity (M = 4.61) to increase more than the share of imported electricity (M = 4.18). Respondents also expect a slight increase in both domestic societal conflicts over energy infrastructure (M = 4.60) and energy-related controversies with neighboring

countries (M = 4.44). The only energy system characteristics that survey participants expect to decrease in the future is the instance of power outages (M = 3.88). Overall, the public expects the largest diversions from the present in the three *TransitionExtent* dimension items that were created using factor analyses (all items scoring >5.5). The scores of the *SystemState* dimension are more diverse, ranging from sharp increases (for example fossil fuel prices) to decreases (i.e. prevalence of power outages).

Our experimental design allows analyzing the sensitivity of these results towards different timeframes and framings. The energy system expectations of the experimental sample (2030 as the reference year, n = 157) are very similar to those in the main sample with the reference year 2050 (see Table 2). In particular, the energy system expectations constituting the *TransitionExtent*, i.e., the three items describing the degree to which a renewable energy transition takes place are almost identical between the reference years 2030 and 2050. The T-test shows that there are statistically significant differences among four variables of the *SystemState* dimension: 1) The share of imported electricity is expected to be higher in 2030 *(M = 4.6)* than in 2050 *(M = 4.18); t(795) = 3.93, p = 0.00.* 2) The prevalence of power outages is expected to be slightly higher than today in 2030 *(M = 4.18)* and slightly lower in 2050 *(M = 3.88); t(795) = 2.75, p = 0.06.* 3) Controversies with neighboring countries over energy-related issues are expected to occur more frequently in 2030 (M = 4.64) than in 2050 (M = 4.44); t(795) = 1.97, p = 0.04. 4) Electricity prices are expected to be higher in 2030 *(M = 5.01)* than in 2050 *(M = 4.80); t(795) = 1.97, p = 0.05.* Not statistically significant are the differences between the per capita use of electricity and energy-related controversies with neighboring countries, which are both also expected to be higher in 2030 than in 2050. In contrast to the timeframe, the framing (realistic vs. idealistic) seems to produce differences in the estimated share of renewables in 2050. The realistic framing ("*Considering economic and political realities, what do you think the share of renewables will be in 2050*?") resulted in a share of 51.9% (SD 36.7). The idealistic framing ("*According to your own values and preferences, how high should the share of renewables be in 2050*?") resulted in a share of 63.1% (SD 23.7). The difference between the two framings is significant t(795) = 4.71, p = 0.00. This exemplifies that while people generally do not differentiate between the years 2030 and 2050, they do differentiate

**Table 2. Energy system expectations for the year 2030 of the experimental sample (N = 157) compared to 2050 main sample (N = 640).**

| Expectation | *M* | *SD* | *Δ2050* | *t* | *p* |
|---|---|---|---|---|---|
| TransitionExtent items | | | | | |
| Renewables | 5.59 | 1.10 | 0.00 | 0.00 | 1.00 |
| Energy efficiency | 5.62 | 0.95 | 0.01 | 0.10 | 9.19 |
| Electric vehicles | 5.54 | 1.00 | 0.00 | 0.00 | 1.00 |
| SystemState items | | | | | |
| Electricity use per capita | 4.52 | 1.31 | 0.09 | 0.76 | .446 |
| Oil and gas prices | 5.26 | 1.34 | 0.05 | -0.43 | .667 |
| Electricity prices | 5.01 | 1.14 | -0.21 | -1.97 | .049* |
| Imported electricity | 4.60 | 1.20 | -0.42 | -3.93 | .000* |
| Power outages | 4.18 | 1.25 | -0.30 | -2.75 | .060 |
| Societal conflicts over energy infrastructure | 4.62 | 1.14 | -0.02 | -0.19 | .850 |
| Energy related controversies with neighboring countries | 4.64 | 1.06 | -0.20 | -2.07 | .039* |

Notes. Overview of energy system expectation of the subsample (*N* = 157) for the year 2030 with Delta and T-test comparisons to the main sample's 2050 expectations.
*M* = Mean, *SD* = Standard Deviation, Δ2050 = Difference between *M*2050 and *M*2030, *t* = T-Test. *p* = significance,
*p≤.05.

between idealistic preferences and realistic expectations in their responses regarding the future energy system.

## 3.2 Four distinct energy system expectation clusters

The hierarchical cluster analysis resulted in four energy system expectation clusters. We start by presenting the ratings of the clusters for the ten expectations (see Table 3). Then, we present ANOVA results comparing the clusters to other items of the questionnaire. Overall, there are only very few socio-demographic differences between the clusters. Gender, age, educational level, household income or political orientation on a left-right scale are for example not significantly different among the clusters. Most differences are in the acceptance of energy technologies, the support for the national energy strategy, trust in parliament, the energy minister and science, as well as the participants' future orientation, of which we present the most relevant items. Comprehensive tables covering all questionnaire items are provided in the S1 Appendix.

**3.2.1 Cluster 1.** This cluster contains people that tend to expect a transition towards a sustainable energy system with much higher shares of renewable energy (M = 6.31), vastly improved efficiency of appliances and processes (M = 6.12) and much higher shares of electric vehicles (M = 6.21) than today. It is the only cluster with values 6 in all of the *TransitionExtent* expectation variables, which is significantly different form all other clusters. Moreover, this is the only cluster expecting the per capita electricity consumption to decrease in the future (M = 3.23). Consequently, they expect the prices of fossil fuels (M = 5.45) to increase much more than the prices of electricity (M = 4.39) and expect a decrease in electricity imports (M = 3.48). Overall, this cluster expects that the energy transition will be positively associated as the prevalence of power outages (M = 3.31), as well as societal conflicts over energy infrastructure (M = 3.95) and energy-related controversies with neighboring countries are expected to decrease (M = 3.78). Similar to the extent of the energy transition, it is thus also the cluster most expecting the challenges related to the energy transition to be resolvable.

Cluster 1 also perceives the highest need for an energy transition (M = 5.65) among all clusters. In addition, the acceptance of renewable energy technologies (solar, wind, hydro) is significantly higher than in the other clusters. In contrast, nuclear energy is much less acceptable to this cluster than to any other. Consequently, this cluster also entails the highest share of people supporting ES2050 (41% voted yes) and the lowest share rejecting it (8% voted no).

Cluster 1 is the only cluster that is predominantly female (55%) and entails the respondents with the highest consideration of future consequences score and the lowest share of access to a car in the household (see S1 Appendix). Trust in the energy minister and parliament are relatively high and trust in science as well as the self-assessed familiarity with Swiss politics are the highest of all clusters.

**3.2.2 Cluster 2.** Cluster 2 is the biggest cluster in the sample (N = 200). While this group of respondents expects the energy transition to happen (M> = 5.5 in all *TransitionExtent* expectations), they expect it to be accompanied by a range of problematic developments. Most importantly, this group is characterized by the expectation that conflicts both within society over energy infrastructure (M = 5.51) as well as controversies with neighboring countries over energy related issues (M = 5.23) will increase strongly, which is significantly different from all other clusters. A reason for the expectation of increasing international energy-related controversies could lie in the expectation of an increasing need to import electricity (M = 4.83), which is the highest of all clusters. Fear of electricity shortages could also be the reason why

**Table 3. Four energy system expectation clusters with key socio-demographics and items with significant differences such as trust, future orientation and political orientation (N = 640).**

| | Cluster 1 (N = 137) | | Cluster 2 (N = 200) | | Cluster 3 (N = 122) | | Cluster 4 (N = 181) | | Overall cluster difference | |
|---|---|---|---|---|---|---|---|---|---|---|
| | M | SD | M | SD | M | SD | M | SD | F | p |
| **TransitionExtent items** | | | | | | | | | | |
| Renewables | 6.31[b,c,d] | .68 | 5.62[a,c] | .95 | 4.47[a,b,d] | 1.26 | 5.68[a,c] | .88 | 83.65 | .000 |
| Energy efficiency | 6.12[b,c] | .84 | 5.73[a,c] | .87 | 4.42[a,b,d] | 1.37 | 5.95[c] | .77 | 84.58 | .000 |
| Electric vehicles | 6.21[b,c,d] | .79 | 5.52[a,c] | 1.01 | 4.48[a,b,d] | 1.12 | 5.76[a,c] | 1.07 | 68.64 | .000 |
| **SystemState items** | | | | | | | | | | |
| Electricity use per capita | 3.23[b,c,d] | .99 | 4.99[a,c,d] | 1.15 | 4.48[a,b,d] | 1.15 | 5.34[a,b,c] | 1.07 | 111.28 | .000[2] |
| Oil and gas prices | 5.45[c,d] | 1.28 | 5.68[c,d] | .94 | 4.97[a,b] | 1.02 | 4.67[a,b] | 1.55 | 24.58 | .000 |
| Electricity prices | 4.39[b,c] | 1.20 | 5.46[a,c,d] | .98 | 4.79[a,b,d] | 1.02 | 4.40[b,c] | 1.26 | 37.33 | .000 |
| Imported electricity | 3.48[b,c,d] | 1.25 | 4.83[a,c,d] | 1.06 | 4.24[a,b] | .96 | 3.95[a,b] | 1.09 | 44.40 | .000 |
| Power outages | 3.31[b,d] | 1.05 | 4.52[a,c,d] | 1.14 | 3.52[b] | 1.13 | 3.85[a,b] | 1.18 | 36.54 | .000[2] |
| Societal conflicts over energy infrastructure | 3.95[b,d] | 1.21 | 5.52[a,c,d] | .91 | 4.25[b] | .95 | 4.33[a,b] | 1.03 | 79.66 | .000[2] |
| Energy related controversies with neighboring countries | 3.78[b,d] | 1.09 | 5.23[a,c,d] | .93 | 4.02[b,d] | .90 | 4.35[a,b,c] | .83 | 78.98 | .000 |
| **Socio-demographics** | | | | | | | | | | |
| Women (N = 639) | .55 | .50 | .47 | .50 | .47 | .50 | .49 | .50 | .79 | .499[1] |
| Age (in years) | 45.33 | 15.1 | 46.72 | 15.0 | 43.65 | 14.5 | 43.13 | 15.4 | 2.12 | .097 |
| CFC 12-pt. (higher implies more future orientation) | 58.6[b,c,d] | 7.44 | 55.7[a,c] | 7.80 | 52.2[a,b,c] | 7.00 | 55.0[a, c] | 7.87 | 15.80 | .000 |
| **Political orientation and trust** | | | | | | | | | | |
| Left/right leaning on the political scale (5 pt.) | 2.86[b,c] | .99 | 3.20[a] | .94 | 3.18[a] | .92 | 3.05 | .99 | 3.84 | .010 |
| Self-assessed familiarity with CH politics | 5.73 | 1.57 | 5.64[c] | 1.72 | 4.81[b] | 1.79 | 5.36 | 1.93 | 7.30 | .000 |
| Belief in value of voting (My vote makes a difference) | 4.36[c] | 1.78 | 4.04 | 1.63 | 3.72[a,d] | 1.64 | 4.28[c] | 1.73 | 3.86 | .009 |
| Trust in parliament | 4.27[c] | 1.51 | 3.88[d] | 1.44 | 3.69[a,d] | 1.46 | 4.34[b,c] | 1.36 | 7.07 | .000 |
| Trust in energy minister | 4.10 | 1.71 | 3.62[d] | 1.70 | 3.62[d] | 1.49 | 4.20[b,c] | 1.65 | 5.73 | .001 |
| Trust in science | 5.32[b,c] | 1.19 | 4.89[a,c] | 1.28 | 4.18[a,b,d] | 1.43 | 5.05[c] | 1.23 | 18.71 | .000 |
| **Energy attitudes** | | | | | | | | | | |
| Perceived need of an energy transition | 5.65[b,c,d] | 1.46 | 5.09[a,c] | 1.48 | 4.53[a,b,d] | 1.46 | 5.14[a,c] | 1.40 | 12.89 | .000 |
| Preference for locally produced electricity | 4.80[c] | 1.64 | 4.74[c] | 1.53 | 4.09[a,b,d] | 1.54 | 4.69[c] | 1.45 | 6.01 | .000 |
| Support for Photovoltaics | 6.49[b,c,d] | 1.01 | 6.01[a,c] | 1.06 | 5.12[a,b,d] | 1.57 | 6.10[a,c] | 1.10 | 30.68 | .000[2] |
| Support for nuclear power | 1.94[b,c,d] | 1.35 | 2.79[a] | 1.86 | 2.90[a] | 1.64 | 2.62[a] | 1.59 | 9.54 | .000[2] |
| Support for natural gas | 3.28 | 1.58 | 3.28 | 1.58 | 3.48 | 1.54 | 3.45 | 1.50 | .42 | .742 |
| ES2050 yes (N = 191) | .41[b,c] | .49 | .26[a] | .44 | .20[a] | .41 | .33 | .47 | 5.29 | .001[1,2] |
| ES2050 no (N = 100) | .08[b] | .27 | .22[a] | .41 | .16 | .37 | .14 | .35 | 3.88 | .009[1,2] |

*Notes.* *M* = mean, *SD* = standard deviation. *F* = variance of the group means, *p* = significance. One-way ANOVA was performed to identify significant differences among the clusters. Bonferroni corrections were used for post-hoc analysis.

[1] The dichotomous variables were tested with chi-square.

[2] Levens homogeneity of variance is significant, which is why Games-Howell post-hoc corrections were applied.

[a] cluster is significantly different from cluster 1 (p≤.05) l.

[b] cluster is significantly different from cluster 2 (p≤.05).

[c] cluster is significantly different from cluster 3 (p≤.05).

[d] cluster is significantly different from cluster 4 (p≤.05).

this is the only cluster expecting an increase in power outages (M = 4.52). The pessimistic view on the energy transition is complemented by the expectation of a significant increase in per capita electricity consumption (M = 4.99) as well as the highest prices for both electricity (M = 5.46) and fossil fuels (M = 5.68).

People belonging to this cluster were most likely to reject ES2050 (22% voted no) despite having high scores in the need for an energy transition and the preference for locally produced electricity. Moreover, renewable energy sources are perceived almost as positively as by Cluster 1. In contrast, Cluster 2 perceives nuclear power significantly more positive than all other clusters. This is the oldest (M = 46.7 years) of the clusters and has rather low trust in general, particularly in the energy minister.

**3.2.3 Cluster 3.** Cluster 3 expects only small divergences from the present throughout all energy system expectations. For example, it expects only slight increases in the share of renewables, electric cars or the efficiency of appliances (M≥4.5). As these changes are expected to be minor, also the respective impacts on society or international relations are expected to be small. The biggest divergence from the present this cluster expects is in the price increase for electricity (M = 4.79) and fossil fuels (M = 4.97).

One quarter of respondents belonging to this group did not vote on ES2050, the highest share among all clusters (see S1 Appendix), while those who voted were divided (20%yes, 16% no). Similar to the energy system expectation, this cluster's energy attitudes tend not to diverge much from the "*Neither agree nor disagree*" option. Exceptions are the dislike of nuclear power, which is in line with the other clusters, and their relatively high acceptance of electricity imports.

Compared to other clusters, it is rather uninterested in energy topics and is characterized by a passiveness in political engagement (see S1 Appendix). They have the lowest values of all cluster for the trust in parliament, the energy minister and science and the lowest consideration of future consequences.

**3.2.4 Cluster 4.** Cluster 4's expectation patterns mostly fall between cluster 1 and 2. The key differences characterizing Cluster 4 are their expectation for a massive increase in per capita electricity consumption (M = 5.34, significantly the highest score of all clusters) and their simultaneous expectation of low energy prices for both electricity (M = 4.40) and fossil fuels (M = 4.67).

Cluster 4 is the second largest supporter of ES2050 (33% voting yes). For nearly all energy attitudes, their scores are between Cluster 1 and Cluster 2, i.e. favorable towards renewable and locally produced electricity. Notable is the highest acceptance of deep geothermal energy of all clusters (M = 4.47).

This is the youngest of all clusters (M = 43.13 years), with the highest average level of education, access to a car in the household (84%), and the lowest share of homeowners. Levels of trust in parliament, the energy minister and science are high.

## 3.3 Comparison of expectations with projections of techno-economic energy scenarios

This section presents the scenario projections from the policy-relevant scenario "*Energy Perspectives*" that correspond with the energy system expectations and describes their fit with the four clusters. Cluster 1 is most closely aligned with the scenario "*Energy Perspectives*" (see Table 4). The only three expectations with only an average fit with the corresponding scenario projection are the share of electricity imports (which the cluster expects to decrease and the scenario projects the share to remain at today's level), electricity prices (which the cluster expects to increase less than the scenario) and power outages (which the cluster expects to decrease and the scenario again projects the share to remain at today's level). Cluster 1 is the only cluster where all *TransitionExtent* expectations are rated to have a close fit (massive increase in renewables, electric vehicles and energy efficiency) with the scenario. Furthermore, all other clusters expect an increase in per capita electricity consumption which is why only

**Table 4. Rated fit of the four cluster's energy system expectations with the corresponding projection from the policy-relevant scenario "Energy Perspectives".**

| Expectation for 2050 | Energy scenario projection for 2050 | Cluster 1 fit | Cluster 2 fit | Cluster 3 fit | Cluster 4 fit |
|---|---|---|---|---|---|
| TransitionExtent items | | | | | |
| Renewables | From 1.38 TW/h in 2010 to 24 TW/h (excluding hydropower). | Close (M = 6.31) | Average (M = 5.62) | Distant (M = 4.47) | Average (M = 5.68) |
| Energy efficiency | Varying across appliances and sectors, but very significant efficiency gains are assumed overall. | Close (M = 6.12) | Close (M = 5.73) | Distant (M = 4.42) | Close (M = 5.95) |
| Electric vehicles | From 0.03% in 2010 to 41%. | Close (M = 6.21) | Average (M = 5.52) | Distant (M = 4.48) | Average (M = 5.76) |
| SystemState items | | | | | |
| Electricity use per capita | Minus 10% compared to 2010. | Close (M = 3.23) | Distant (M = 4.99) | Distant (M = 4.48) | Distant (M = 5.34) |
| Oil and gas prices | Plus 100% compared to 2010. | Close (M = 5.45) | Close (M = 5.68) | Average (M = 4.97) | Average (M = 4.67) |
| Electricity prices | Plus 42% compared to 2010. | Average (M = 4.39) | Close (M = 5.46) | Close (M = 4.79) | Average (M = 4.40) |
| Imported electricity | Larger variance throughout the year (importing during winter, exporting during summer), but stable overall. | Average (M = 3.48) | Average (M = 4.83) | Close (M = 4.24) | Close (M = 3.95) |
| Power outages | A highly reliable electricity system is implicitly assumed. | Average (M = 3.31) | Average (M = 4.52) | Close (M = 3.52) | Close (M = 3.85) |
| Societal conflicts over energy infrastructure | Social acceptance and cohesion is implicitly assumed as the whole strategy is considered to be feasible. | Close (M = 3.95) | Distant (M = 5.52) | Close (M = 4.25) | Close (M = 4.33) |
| Energy related controversies with neighboring countries | Implicitly regarded to be non-existent, energy imports assumed to be available at all times. | Close (M = 3.78) | Distant (M = 5.23) | Close (M = 4.02) | Close (M = 4.35) |

Note: Fit between the scenario projection and the public's expectation as rated by the authors. Expectations rated to have a close fit to the corresponding scenario projection are shaded green. Expectations rated to have a average fit to the corresponding scenario projection are shaded grey. Expectations rated to have a distant fit to the corresponding scenario projection are shaded red. M = mean.

the expected decrease of Cluster 1 has a close fit with the scenario projection. Cluster 3 was rated to have a distant fit with the scenario projection on three occasions. Besides the electricity use per capita, it concerns the expected increase in societal and international conflicts, which is not projected by the scenario. Cluster 3 had a distant fit on four occasions. This relates to all of the *TransitionExtent* expectations (cluster expects a persistence of the status quo) and to the electricity use per capita.

## Discussion

### 4.1 Public energy system expectations illustrate the pervasiveness of the energy transition as an idea

The first research question of this paper asked what the public's expectations about the techno-economic aspects of the energy system are. The results suggest that the public does expect the energy system to change significantly in the future. The fact that this is also true for individuals who are critical of the Swiss energy policy indicates that the fact that a transition of some sort will take place is a widely shared and deeply rooted belief among Swiss citizens. This is remarkable because people typically tend to underestimate changes that happen over a long timescale, especially in large socio-technical systems that have been functioning and stable for decades [45]. Hence, the assessed expectations indicate "*a psychological readiness to engage in the transition [. . .]*" that Vainio et al. [23] also attested to their sample in a survey assessing citizens' images of a sustainable energy transition.

The variance among the expectations of the main sample and the comparison between the main sample and the experimental sample provide insights for the interpretation of these expectations. First, the significant differences between the realistic and the idealistic framing in participants' estimation of the future share of renewables confirmed the importance of framings in attitude surveys, as it has been previously highlighted by Clarke et al. [46]. Yet, we found only a few differences between the energy system expectations for the year 2050 (main sample) and the year 2030 (experimental sample). This indicates that public energy system expectations are conceptually different from scenario projections [23]. Particularly, expectations tend to be static in the sense that they do not describe a path-to-the-end state, but rather the future end state itself. This is evident in the increased cost of fossil fuel prices and the number of electric vehicles in the passenger car fleet that often only begin to rise significantly after 2030 in energy scenarios, but are nearly identical for the time horizons 2030 and 2050 in the public expectations. As there are not many significant differences between the 2030 and the 2050 time horizons, one can question whether people differentiate between the two or whether both are perceived to be distant futures. However, there were differences in the electricity prices, the frequency of power outages, and the risk of controversies with neighboring countries over energy-related issues, which are all expected to be significantly higher in 2030 than in 2050. The expected energy future in 2050 as a whole is thus viewed more positively than the energy future in 2030 [47].

Public energy system expectations mirror the key promises and concerns associated with the energy transition [29]. Increasing energy costs and societal conflicts are, for example, clearly among the most common concerns among the expectations. However, one characteristic that is controversial and prominent both in academic literature and the political campaigns surrounding ES2050 in Switzerland, but not reflected in public expectations, is energy security [33]. The majority of respondents neither expect reliance on foreign electricity sources to increase in the future, nor power outages to become more widespread. In fact, the main sample expects a further decrease in power outages by 2050, which is astonishing considering Switzerland only experienced a cumulative total of 20 minutes without power in 2017, ten of which were due to unforeseen circumstances [48]. This suggests an expert/non-expert divide which future research could use as an interesting case to advance the understanding of how expectations influence how people integrate new and sometimes contrasting information [14]. That experts and non-experts can have different preferences for the future energy technology mix in Switzerland has recently been demonstrated by Xexakis et al. [15].

## 4.2 Relationship between expectation clusters and the acceptability of a sustainable energy transition

The variance between the clusters suggests that within the population there exist very different expectations about the energy future. Moreover, the clusters represent four different conceptualizations of the energy future consisting of distinct combinations of promises and concerns. We argue that these conceptualizations are not arbitrary. Cluster 1 focuses on the potential benefits associated with the energy transition and the respective respondents can thus be considered transition optimists. Cluster 2, in contrast, focuses on the potential risks associated with energy transitions and can thus be labelled transition pessimists. At the same time, Cluster 2 acknowledges the need for an energy transition and is not per se against renewable energy, indicating a certain ambivalence. Cluster 3 is the only one that expects the whole set of energy system characteristics to remain stable. The reason for the belief that the status quo will remain far into the future could correspond with this cluster's indifference about energy topics and their low self-assessed knowledge and activity in political processes. The rationality of

Cluster 4 is defined by the assumption that there will be an abundance of various energy sources in the future. Interestingly, this cluster expects that there will be a transition towards renewable energy sources, but at the same time expects this to happen without large increases in the prices for fossil fuels.

We do not claim that these expectations a comprehensive operationalization of the complexities and interdependencies of energy systems or that they are in line with expert views on the energy future. In fact, section 4.3 shows that there are some major deviations from the formalized expert projections of the reference energy scenarios of the ES2050. While the deviations differ among the respective clusters, all clusters follow a certain logic that allows for inferring the key ideas of the energy future shaping the expectations. The clusters seem to align with the support for Switzerland's national energy strategy ES2050. There are several other significant relationships between the clusters and their related energy technology preferences and attitudes towards energy policies. However, socio-demographic differences between the clusters were less clear and seem to be of minor importance. This contrast with a lot of acceptance research on energy technologies and policies where socio-demographic variables often play a significant role [49].

In contrast, trust seems to be a key concept when it comes to why people associate the energy transition more with potential benefits or risks respectively. Trust in parliament, the energy minister and in science are significantly different between the clusters. In a review article, Huijts et al. [50] show that trust is particularly important as a heuristic when people know little about a topic. As there are many uncertainties associated with energy transitions and the effects and involved actors are manifold it seems logical that "*positive expectations of the intentions or behavior of another*" as trust was defined by Rousseau et al. [51] is critical.

In addition, the hierarchical cluster analysis shows that, depending on the underlying rationality of the energy system expectations, the same promises and concerns can be interpreted differently. For example, for Cluster 1, the anticipated reduction in electricity demand by 2050 seem to reflect a positive step. Possibly, it symbolizes increased efficiency and careful use of energy resources in general. In contrast, for Cluster 4, the anticipated sharp increase in electricity demand seems to be positively associated with a sustainable energy transition. This could be due to the increased degree of electrification and prevalence of "smart" appliances. Hence, the underlying conceptualizations of how an energy transition works and the different opinions about its key target (for example climate change mitigation, energy autarky or decentralization) determine the appraisal of energy technologies or policies [52]. Accordingly, promises and concerns are not universal, but contingent on personal conceptualizations of the energy future [16].

## 4.3 The varying compatibility of energy system expectation clusters and projections of the national energy scenario

The interaction of the public with energy scenarios is not comparable to the scrutiny applied by energy and modeling experts. Nevertheless, the results of this study suggest that scenario-derived promises and concerns circulated by the media and political discussions could nevertheless provide powerful reference points for the energy-related expectations of non-experts, as it has been observed under experimental conditions by Demski et al. [13]. Cluster 1, whose expectations have most expectations that are in line with the national energy scenario, exemplifies this. This cluster has the highest support rate for ES2050 and the most trust in science, indicating that this group could perceive the projections of the scientifically derived energy scenario to be credible.

At the same time, it is evident that most respondents have energy system expectations that differ significantly from the national energy strategy projections on a number of different dimensions. The largest contrast between expectations and scenario was evident in the anticipated electricity demand, which only Cluster 1 expects to decrease in line with the scenario projection. Many people associate energy with progress, which could explain why most people expect an increase in electricity demand [53]. Also, most people's personal experiences and lifestyles (i.e. more and bigger electric appliances, trends towards electrification in many jobs) could iterate the perception of more electricity use, while energy efficiency improvements are typically much less noticeable. However, the fact that most people who expect an increasing electricity demand still support ES2050 shows that the acceptability of a broader policy package is not contingent on particular promises and concerns. In contrast, a holistic view on the public's energy system expectations demonstrates a certain willingness to act or at least accept changes towards the general direction of a renewable energy transition.

Accordingly, there co-exists a range of expectations about the energy system that are more or less compatible with the scenario constituting the national energy strategy. This plurality of distinct energy system expectations could also correspond with the diversity of energy scenarios that exists. However, to date it is largely unclear what determines the uptake of scenarios and how their contested projections of future energy systems are perceived. A study among researchers showed the selection and application of energy scenarios is not determined by the users' field of study, but by the personal background and purpose of scenario use [54]. This tendency was confirmed by a study on the use of climate scenarios which found that a user's sectoral background was not a significant predictor for the type of scenario application [55]. Hence, it can be assumed that the uptake and relevance of scenario projections, for example as distinct promises and concerns proliferated by media and political discussions, is only loosely correlated with the publics' socio-demographic background. The results of this study show that trust, future-orientation and political activity are better predictors for the relationship between personal expectations and formalized scenario projections. It may well be that these attributes in turn correlate with media use patterns and affinity to follow political discussions in general.

The assessed energy system expectations can also make explicit what energy scenarios only consider implicitly, for example as *ceteri paribus* conditions. This includes the occurrence of societal conflicts over energy infrastructure or controversies with neighboring countries over energy related issues. The correlation between the acceptability of renewable energy technologies and the support for ES2050 shows that the reason for Cluster 2 to predominantly vote against the Swiss energy transition lies exactly in these factors that typically are outside the focus of techno-economic energy scenarios. If it is indeed these social factors determining the acceptability of an energy scenario or a corresponding energy strategy, it raises the question how relevant it is to publish energy scenarios with their traditional focus on techno-economic aspects that can be quantified. Can scenarios enable an enlightened energy discourse, as suggested by [56], when the key elements for non-experts to create meaningful and relatable storylines [15] to make sense of the energy future are missing?

## 4.4 Critical reflection and outlook

The study has some limitations. First, it is exploratory in nature, using cluster analysis of a novel set of promises and concerns as proxy for techno-economic energy system expectations. Second, the expectations were assessed over a single time period in a rather confined geographical region. As energy transition are strongly context dependent, generalizations should only be made on the basis of an analysis of the respective situation in other contexts.

Although challenging, it would be particularly interesting for future research to monitor public energy system expectations over a longer time in order to understand the formation and dynamic aspects of expectations. Longitudinal studies could shed further light on the impact of critical events, political cycles or generational effects on the persistence of expectations. For example, the study was conducted before the issue of climate change received a major boost in visibility—inter alia through the climate strike youth. Thus, comparative analyses covering multiple language regions or countries could yield interesting insights into cultural specificities, generational effects and respective expectation patterns. Third, no standardized way of comparing expectations to scenario projections exists to date, hence in this study a direct approach was chosen which worked well for many expectations, but not all. As scenario products are often distorted or simplified when they are communicated to non-expert communities, future research could use discourse analysis to identify the relevant promises and concerns in energy debates. Based on the insights presented in this study, we argue that it is worthwhile to investigate the role of expectations and their interdependence with model-based energy scenarios. As of today, it is not clear whether the public's energy system expectations or energy discourse more generally are actually influenced by scenario projections or whether scenarios basically analyze the techno-economic feasibility of expectations that are deeply rooted in society and thus also among scenario developers.

## Conclusion

Energy transitions are co-evolutionary processes between social groups, their behavioral patterns and technologies. While expert perspectives tend to be well understood, the public understanding of transitions is still not. Assessing energy system expectations could be a first step in this direction. Our study used an exploratory approach to assess public expectations of the techno-economic energy system aspects for the years 2030 and 2050 with separate samples and compared them to the policy-relevant energy scenario projection. It thus provides a first attempt to assess the public's expectations about the energy system in a non-experimental setting.

We identified four clusters of energy system expectations. Each of these describes a distinct and holistic vision of the energy future. We argue that the variance between the clusters does not indicate arbitrariness, but rater variance in how the public perceives the energy future. Cluster 1 is very optimistic about the energy future, while Cluster 2 is generally more pessimistic and particularly worried about energy related conflicts. Cluster 3 is the only cluster not expecting an energy transition at all, indicating that the concept of an energy transition has become a collective expectation shared by a large majority of the public. Cluster 4 expects an increase in electricity demand and a simultaneous reduction in electricity prices, which not only stands in contrast to the expectations of the other three clusters, but also to the projection of the national energy scenario study which defined the Swiss Energy Strategy 2050. These different peculiarities of energy system expectations should be recognized by researchers and decision-makers communicating energy-related topics.

While energy system expectations tend to be static images of the future that vary only very little even between different timeframes, energy scenarios provide highly specific what-if pathways. Our analysis showed that many expectations determining the acceptability of the energy transition are only implicitly represented in energy scenarios. Scenario projections thus miss key aspects the public worries or is hopeful about in relation to the energy future. Accordingly, if the goal of publishing energy scenarios is to increase the transparency of policymaking, the scenario content also needs to be tailored at the public's interests and competencies. For example, while the timing of energy investments and technology developments is a critical aspect in energy scenarios, our analysis showed that most respondents do not differentiate between the

timeframes 2030 and 2050. The strong correlations of the four clusters with the acceptability of energy technologies and support for the national energy strategy indicate that it would be worthwhile to further investigate the interdependencies between public energy system expectations and energy scenarios. Energy system expectations can function as a proxy for the range of energy futures that are attainable according to public perception.

## Supporting information

**S1 Appendix. Contains a sample description, an overview of different cluster solutions, internal consistencies of the energy system expectation scales (using Cronbach's α), the complete ANOVA tables as well as the full questionnaire used (German original and English Translation).** The Appendix contains the following tables: S1 Table Socio-demographic sample description and comparison with Swiss population. S2 Table Overview of cluster solutions. S3 Table Internal consistency of energy system expectation scales (using Cronbach's α). S4 Table Differences between the energy system expectation clusters with respect to socio-demographics, future orientation and political orientation. S5 Table Differences between the energy system expectation clusters with respect to energy attitudes and voting behaviour in the ES2050 referendum. S6 Table Questionnaire with survey items (german and English Translation). (PDF)

## Acknowledgments

The authors thank everyone who took part in the extensive pretest of the survey. Special thanks goes to Aya Kachi, Rebecca Lordan-Perret and Fintan Oeri for survey design and initial data analysis.

## Author Contributions

**Conceptualization:** Lukas Braunreiter, Michael Stauffacher, Yann Benedict Blumer.

**Data curation:** Lukas Braunreiter.

**Formal analysis:** Lukas Braunreiter.

**Methodology:** Lukas Braunreiter, Michael Stauffacher, Yann Benedict Blumer.

**Project administration:** Lukas Braunreiter.

**Supervision:** Michael Stauffacher, Yann Benedict Blumer.

**Visualization:** Lukas Braunreiter, Yann Benedict Blumer.

**Writing – original draft:** Lukas Braunreiter.

**Writing – review & editing:** Michael Stauffacher, Yann Benedict Blumer.

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
