## [Decision Letter · Decision Letter 0]

15 Nov 2019

PONE-D-19-25898

Expecto transitio: Exploring non-experts’ techno-economic expectations of the energy future

PLOS ONE

Dear Mr. Braunreiter,

Thank you for submitting your manuscript to PLOS ONE. After careful consideration, we feel that it has merit but does not fully meet PLOS ONE’s publication criteria as it currently stands. Therefore, we invite you to submit a revised version of the manuscript that addresses the points raised during the review process.

Both reviewers agreed in minor revisions. They made only two major points which I would like to raise here. I would like to ask you to reduce speculations in text as well as in conclusions which will strengthen the scientific quality of the manuscript. In addition I would like to ask you to provide the background data of study following the open access strategy to increase the availability of the data.

We would appreciate receiving your revised manuscript by Dec 30 2019 11:59PM. To enhance the reproducibility of your results, we recommend that if applicable you deposit your laboratory protocols in protocols.io, where a protocol can be assigned its own identifier (DOI) such that it can be cited independently in the future. For instructions see: http://journals.plos.org/plosone/s/submission-guidelines#loc-laboratory-protocols

We look forward to receiving your revised manuscript.

Kind regards,

Bruno Merk

Academic Editor

PLOS ONE

Journal Requirements:

1. 

2.   We noted in your submission details that a portion of your manuscript may have been presented or published elsewhere.   Please clarify whether this [conference proceeding or publication] was peer-reviewed and formally published. If this work was previously peer-reviewed and published, in the cover letter please provide the reason that this work does not constitute dual publication and should be included in the current manuscript.

Additional Editor Comments (if provided):

Both agreed in minor revisions. They made only two major points: reduce speculations in text as well as in conclusions and the availability of the data.

Reviewers' comments:

Reviewer's Responses to Questions

**Comments to the Author**

1. Is the manuscript technically sound, and do the data support the conclusions?

Reviewer #1: Yes

Reviewer #2: Partly

2. Has the statistical analysis been performed appropriately and rigorously? 

Reviewer #1: Yes

Reviewer #2: Yes

3. Have the authors made all data underlying the findings in their manuscript fully available?

Reviewer #1: Yes

Reviewer #2: Yes

4. Is the manuscript presented in an intelligible fashion and written in standard English?

Reviewer #1: Yes

Reviewer #2: Yes

5. Review Comments to the Author

Reviewer #1: I really enjoyed this article and as far as I believe the survey and follow up analyses were sound. I would love to know why you only chose a sample from the Swiss-German speaking community, and how did you select and engage this sample. Also, although not necessary for this paper, it would have been highly insightful to have had a comparative sample from another area of Switzerland. Sometimes the conclusions seem speculative rather than solidly derived from the data - however I understand the nature of survey analysis and that this often is the case. Also I feel that the conclusion should be revised and made a little more obvious about the outcomes/conclusions of the survey. Also, throughout the article and particularly within the conclusion there are some English language errors - easily corrected - but worth revising especially to enhance the meaning of the conclusion.

As far as I understand the survey is available but the actual survey returns are not on the system. I am not sure if this is standard (?).

Reviewer #2: This paper describes an interesting exploratory study on the role of the non-expert public in energy futures and energy transitions, with Switzerland serving as a case study. The use of cluster analysis seemed useful and appropriate. The study findings that there were few differences between the energy systems expectations for 2030 vs. 2050 was especially interesting, though I would caution the authors in general, and in this instance in particular, to not speculate too much on the meaning behind findings such as these that they cannot explain by their analysis.

My main concern about this paper was the very limited information provided about the survey sample, respondents, and its representativeness of the general population. Reference was made to more detailed descriptions of the questionnaire and the participant recruitment process in Blumer et al. 2018, but some of that detail should be described in this paper too. For instance, why was only German speaking populations in the country recruited, and not French speaking populations in Switzerland as well? Did the authors attempt to develop a random or stratified random sample that was representative of the general population, and if not why not? How successful were they in this effort?

The paper was generally well written but I found a few exceptions:

line 296, "increase more s than the share ..." means what?

line 474, "The variance among in the expectations ..." means what?

line 519, "thus considered to be transition optimists" - would it be clearer to say "thus be considered transition optimists." ?

line 562, "Clusters and Projections", not "Clusters ad Projections"

line 608, "I think you mean "factors that typically are outside the", not "factors that typically is outside the"

6. PLOS authors have the option to publish the peer review history of their article (what does this mean?). If published, this will include your full peer review and any attached files.

Reviewer #1: No

Reviewer #2: Yes: Barry D Solomon

---

## [Author Response · Author response to Decision Letter 0]

28 Nov 2019

We respnded to each editor and reviewer comment in the 'Response to the reviewer' file. 

We like to thank the two reviewers as well as the editor for their time and effort to provide us with a very helpful feedback. Apart from pointing out specific mistakes and passages that require clarification, they also motivated us to improve some aspects of the manuscript more systematically. 

In particular, we addressed two main points raised by both reviewers. First, we extended our description of the sampling process and provide more information on how the sample compares to the general population in Switzerland. Second, we revisited the manuscript to achieve a better alignment of discussion and data.

Our answer to each review comment and corresponding actions can be found below. In these answers, all changes that concern the content of our manuscript are reflected. Typos and small mistakes that we spotted during the general overhaul of our submission are not listed individually, but are documented in the track-changes version of the manuscript. The same applies to changes that are associated with an adaption of the manuscript to the formal guidelines of PLOS One, which mainly concerned headings and the title page. 

We feel that the revision process has improved our paper’s quality significantly and are confident that it is now on a level that makes it a relevant and adequate research contribution. These improvements would not have been possible without the valuable and constructive feedback we received. We are extremely grateful for this.

---

## [Editor Report · Decision Letter 1]

18 Dec 2019

Exploring and clustering non-experts’ techno-economic expectations of the energy future

PONE-D-19-25898R1

Dear Dr. Braunreiter,

We are pleased to inform you that your manuscript has been judged scientifically suitable for publication and will be formally accepted for publication once it complies with all outstanding technical requirements.

With kind regards,

Bruno Merk

Academic Editor

PLOS ONE

Additional Editor Comments (optional):

Congratulations!
---

## [Editor Report · Acceptance letter]

5 Feb 2020

PONE-D-19-25898R1 

Exploring and clustering non-experts’ techno-economic expectations of the energy future 

Dear Dr. Braunreiter:

I am pleased to inform you that your manuscript has been deemed suitable for publication in PLOS ONE. Congratulations! Your manuscript is now with our production department. 

With kind regards,

on behalf of

Prof. Dr. Bruno Merk 

Academic Editor

PLOS ONE